# Iron in Alzheimer’s Disease: From Physiology to Disease Disabilities

**DOI:** 10.3390/biom12091248

**Published:** 2022-09-06

**Authors:** Amit Pal, Giselle Cerchiaro, Isha Rani, Mariacarla Ventriglia, Mauro Rongioletti, Antonio Longobardi, Rosanna Squitti

**Affiliations:** 1Department of Biochemistry, All India Institute of Medical Sciences (AIIMS), Kalyani 741245, West Bengal, India; 2Center for Natural Sciences and Humanities, Federal University of ABC (UFABC), Avenida dos Estados, 5001, Bl.B, Santo André 09210-580, SP, Brazil; 3Department of Biochemistry, Maharishi Markandeshwar University (MMU), Mullana, Ambala 133203, Haryana, India; 4Fatebenefratelli Foundation for Health Research and Education, AFaR Division, 00186 Rome, Italy; 5Department of Laboratory Medicine, Research and Development Division, Fatebenefratelli Isola Tiberina, Gemelli Isola, 00186 Rome, Italy; 6Molecular Markers Laboratory, IRCCS Istituto Centro San Giovanni di Dio Fatebenefratelli, 25125 Brescia, Italy

**Keywords:** Alzheimer’s disease, ferroptosis, iron, ceruloplasmin, reactive oxygen species

## Abstract

Reactive oxygen species (ROS) play a key role in the neurodegeneration processes. Increased oxidative stress damages lipids, proteins, and nucleic acids in brain tissue, and it is tied to the loss of biometal homeostasis. For this reason, attention has been focused on transition metals involved in several biochemical reactions producing ROS. Even though a bulk of evidence has uncovered the role of metals in the generation of the toxic pathways at the base of Alzheimer’s disease (AD), this matter has been sidelined by the advent of the Amyloid Cascade Hypothesis. However, the link between metals and AD has been investigated in the last two decades, focusing on their local accumulation in brain areas known to be critical for AD. Recent evidence revealed a relation between iron and AD, particularly in relation to its capacity to increase the risk of the disease through ferroptosis. In this review, we briefly summarize the major points characterizing the function of iron in our body and highlight why, even though it is essential for our life, we have to monitor its dysfunction, particularly if we want to control our risk of AD.

## 1. Introduction

Alzheimer’s disease (AD) is the most common form of dementia, defined as a group of symptoms [1]. Major neurodegenerative dementias are a significant unmet medical burden and a growing global health challenge. Dementia afflicts about 55 million individuals worldwide, with almost 10 million new cases each year (WHO 2021). AD impacts thinking, memory, and independence. It represents 60–80% of all types of dementia (Alzheimer’s Association. 2021 Alzheimer’s Disease Facts and Figures) and is typified by extracellular amyloid-β (Aβ) peptide formations that create the amyloid plaques and by phosphorylated tau protein (P-tau) accumulation making neurofibrillary tangles [2,3]. In 2019, people suffering from AD had been estimated at 50 million [4]. Based on the onset of clinical symptoms, AD can be referred to as early- or late-onset (<65 years old, or >65 years old, respectively) and it is also distinguished in familial (<5%), and sporadic, with a multifactorial complex disease etiology. The early-onset familial AD is identified in the presence of mutations in *Amyloid Precursor Protein* (*APP), Presenilin1 (PSEN1)*, or *Presenilin2 (PSEN2)* genes [5].

Old age, having relatives with AD, and genetics (*APOE4* gene encoding the apolipoprotein E4) are the most significant risk factors for late-onset AD [1]. A gradual loss of cognition characterizes the disease course, eventually leading to dementia and death. As per the 2018 National Institute on Aging—Alzheimer’s Association (NIA-AA) Research Framework [6], a syndromal cognitive staging scheme including cognitive unimpaired (normalcy), mild cognitive impairment (MCI) and dementia can be distinguished. The final stage of dementia can be further divided into mild, moderate, and severe stages. The International Working Group (IWG) and the US National Institute on Aging–Alzheimer’s Association in 2014 defined new criteria for AD diagnosis [7] based on recent advances in neuroimaging and biomarkers including: increased tracer retention on amyloid positron emission tomography (PET) or decreased Aβ 1–42 together with increased total-tau (T-tau) or phosphorylated (P-tau) in cerebrospinal fluid (CSF). The hallmark of AD are deposits of amyloid plaques and neurofibrillary tangles in the brain. Neurofibrillary tangles are aggregates of the modified protein tau in the intracellular space that undergo hyper-phosphorylation (P-tau), which causes the proteins to aggregate and break down neuron architecture.

Amyloid plaques are aggregates of protein fibrils that form insoluble aggregates in the extracellular space, around neurons or glial cells. Their core is mainly made of Aβ, a 4 kDa peptide of 39 to 43 amino acids, derived from the APP, a transmembrane protein that is highly expressed at cerebral cortex synapses. APP is cut on the extracellular side by the enzyme α-secretase and γ-secretase, but it can also be cleaved by β-secretase resulting in the Aβ peptide release in the extracellular space, which then aggregates in senile plaques [8].

In the last two decades, it has become well established that oxidative stress affects brain functionality. This line of research has uncovered the vital role of transition metals involved in several biochemical reactions producing free radicals. Iron and copper are well known to take part in Fenton-type reactions which generate reactive oxygen species (ROS) that damage and destroy cellular compartments [9]. A bulk of evidence has uncovered the role of metals in the generation of the toxic pathways at the base of AD. However, metals, as potential drivers of AD, have been side-lined starting from the 1990s. The advent of the Amyloid Cascade Hypothesis [8,10] obscured all the other research fronts, albeit potentially explanatory of some of the most important deficits or abnormalities that characterize at least subsets of AD patients [11]. Herein, we will discuss iron involvement in AD, with the aim of providing the reader with a concise description of the fundamental role played by the metal in the pathogenesis of AD—and in particular by its’ cross talking’—with copper and with a logic summary of the reasons urging researches to monitor its homeostasis in relation to the risk of AD.

## 2. Essentiality of Iron

Iron is a d-block transition metal typified by Fe^2+^ (divalent ferrous) and Fe^3+^ (trivalent ferric) oxidation states. It is the most abundant transition metal present in the body and participates in several functions that are essential for our life. Iron is a constituent of hemoglobin (which is important for supplying oxygen to cells), myoglobin, and cytochromes. Myoglobin delivers oxygen to muscle tissue and stores it there; furthermore, iron is involved in catalases’ and peroxidases’ function encompassing in oxygen metabolism, as well as of cytochromes in the mitochondrial respiration chain and electron transport. Fe is involved in myelin and neurotransmitters biosynthesis; it participates in the immune system and DNA synthesis and gene regulation. This metal is essential even though it is poorly bioavailable. Furthermore, it is potentially toxic. As a consequence, cells have evolved high complex mechanisms to handle iron absorption. Hepcidin is a peptide hormone that controls dietary iron absorption and systemic iron traffic in mammals by modulating the levels of ferroportin on the cell surface which facilitates iron efflux from the cell. At cell level, iron uptake, storage, and utilization are regulated by the iron responsive element/iron regulatory protein (IRE/IRP). A concise description of the more relevant regulatory pathways operating at the systemic and cellular levels is reported herein (for specialized review, refer to [12]).

AD is linked to iron proteins, hepcidin, ferritin, and pro-inflammatory biomarkers such as interleukin-1β (IL-1β), IL-6, and ApoE ɛ4 allele. Hepcidin, ferritin, and IL-6 participate significantly in host defense mechanisms associated with neuroinflammation in AD. Hepcidin was found elevated in AD serum patients, indicating its correlation with the disease [13]. 

### 2.1. Iron Absorption and Distribution

There is no physiological mechanism in the human organism for expelling iron, which is minimally eliminated by the death of enterocytes, or by peeling of the skin, or in the case of small bleeding, such as during menstruation in women.

A common Western diet provides a quantity of iron equal to about 15 mg/day, but of this quantity our body absorbs only 1–2 mg/day. That is the quantity it needs [14]. The human body always contains 3–5 g of iron of which 2.5 g are present in hemoglobin, about 600 mg in reticuloendothelial macrophages, another 300 mg are used by proteins to carry out certain cellular processes, 3–4 mg bind to transferrin, and the remainder is stored as ferritin [15].

Serum ferritin is an iron storage protein and has large variations from individual to individual. Only major deficiencies have effects on physiology. Ferritin is high in people with hemochromatosis and other excess iron storage disorders. Ferritin is an acute phase reactant and can be increased in inflammation states: ferritin sequesters iron and inhibits microbial iron scavenging. Its synthesis is cytokine responsive and regulated at both the transcriptional and translational level primarily in hepatocytes and macrophages. It is regulated by TNF-α and IL-1α and by IRP1 or IRP2 [16]. 

In general, circulating iron is bound to transferrin. Its levels are controlled by hepcidin. Hepcidin is a hormone that is biosynthesized in the liver and controls the degradation rate of ferroportin in: (1) the enterocyte membrane, and consequently, the rate of iron export from the enterocyte to the blood; (2) the hepatocytes, to control the transport stored iron; and (3) macrophages, to control the transport recycled iron, and, that as a whole, control systemic iron homeostasis.

The body uses clever reduction and oxidation processes to regulate iron absorption. Virtually all dietary iron is in the ferric oxidation state (Fe^3+^) and must be first reduced to Fe^2+^ to be considered for transport through the apical surface of the enterocytes in the duodenum. In fact, the divalent metal transporter 1 (DMT-1), which binds only divalent iron, facilitates Fe^2+^ transport [17].

Various ferric-reductases are expressed on the enterocyte’s apical surface, such as the Cytochrome b558 ferric/cupric reductase [18] and Steap2 [19] that perform this reduction. In addition, Prion protein (PrPC) has been reported to function as a ferrireductase. It seems to facilitate iron transport across duodenal enterocytes, as shown in PrP^KO^ mice whose systemic iron homeostasis is abnormal. Specifically, a functional role of PrP in iron metabolism has been reported, which might explain the imbalance of iron homeostasis in prion disease as a result of a loss of PrP function [20]. After reduction, DMT-1 channels Fe^2+^ into the enterocyte thanks to an electrochemical gradient of H^+^ (from outside to inside of the cell) [17,21] (Table 1, Figure 1).

There is also a secondary mechanism, not completely understood, by which iron primarily derived from meats enters the enterocytes as part of a heme, so as divalent iron, through the heme carrier protein 1 (HCP-1), which is also present on the enterocyte’s apical surface. At this point, the iron is released by the action of a heme oxygenase and joins the Fe^2+^ pool delivered by DMT1 [22] (Table 1, Figure 1).

Once inside the enterocyte, iron undergoes further absorption control processes. Any iron ‘in excess’ inside the cell is stored into ferritin, a huge 450 kDa protein that can store up to 4500 atoms of iron, where it will remain until the death of the enterocyte when it is recycled or partially secreted through the microvilli [23]. A controlled amount of iron travels instead to the enterocyte’s basolateral surface, which is transferred by ferroportin to the portal plasma [24,25,26]. This is the only portion of iron absorbed. On the basolateral outer surface, the ferroxidase hephaestin re-oxidizes Fe^2+^ into Fe^3+^, which is the only iron form capable of binding transferrin and be finally taken into circulation [27] (Table 1, Figure 1).

A key regulating role is played by the peptide hepcidin (for an exhaustive review, see [28]). When the liver detects an excess of the transferrin-bound iron, it responds by increasing the synthesis of hepcidin, which binds to ferroportin on the enterocytes’ basolateral surface, causing its internalization into the cell, where ferroportin is rapidly degraded [29]. This causes more iron to remain inside the enterocyte and stored into ferritin, thus reducing the amount of iron available for transport by transferrin and released in general circulation. Decreased levels of ferritin (hypoferritinemia and low iron levels) can also occur during inflammation states linked to inflammation-driven increases in hepcidin concentrations [30]. Hepcidin acts to decrease the absorption and availability of iron, despite acute phase increases in iron-binding proteins, such as ferritin that results in a functional iron deficiency. States of iron deficiency are also characterized by low iron levels, high transferrin/iron-binding capacity (TIBC), and low % transferrin saturation. In fact, transferrin is a negative acute phase protein that decreases during inflammation.

Human homeostatic iron regulator protein (HFE) is a protein within the membrane of diverse cell types including enterocytes that controls the absorption of iron by controlling the affinity of TfRs on cell membranes. Mutations in the *HFE* gene cause *HFE* hereditary hemochromatosis by facilitating the absorption of iron from the diet and its consequent over-deposition in the liver, brain, and other organs. Hemochromatosis has a mendelian transmission. It is inherited as an autosomal recessive trait that is associated with the major histocompatibility complex, located on chromosome 6 [31].

Once outside the enterocyte and re-oxidized into Fe^3+^, iron binds transferrin, an 80 kDa glycoprotein synthesized by the liver, by which it is transported in circulation. Transferrin provides a further regulation step. Each transferrin protein binds two Fe^3+^ ions (holo-transferrin) with an affinity that is very high but also pH-dependent [32]. The liver’s transferrin production is adjusted to bind and transport only the percentage of absorbed iron needed by the body. When special conditions arise, the liver modulates transferrin synthesis. For example, in situations of iron deficiency, the liver produces more transferrin, increasing the blood’s total TIBC. On the contrary, when iron is high, such as in hemochromatosis, the liver reduces the production of transferrin to reduce the blood’s TIBC.

An effective way to evaluate the percentage of bound iron is by measuring the so-called transferrin saturation (TSAT), which expresses the ratio of the amount of iron present in the serum to TIBC (multiplied by 100). Measuring TIBC is easier than measuring transferrin directly and provides an accurate measure of the latter. Normal transferrin saturation ranges are 20–45%. Transferrin is also an upstream regulator of hepcidin [33]. For an extensive review on the role of transferrin in iron metabolism see [34].

The majority of absorbed iron is immediately taken by transferrin to the bone marrow to participate in erythropoiesis by building hemoglobin, the oxygen transport protein, and a constituent of red blood cells.

### 2.2. Iron Metabolic Control Unit: The Hepatocyte

Iron not involved in erythropoiesis reaches the liver to join hepatocytes and reticulo-endothelial cells that allow the recycling of iron by ingesting the old red blood cells, breaking down their hemoglobin and releasing the iron content that will be reused [35].

The uptake of iron by hepatocytes occurs through a receptor-mediated endocytosis. This mechanism occurs when a transferrin 1 receptor (TfR1) located on the cell surface [36,37] binds to a holotransferrin, for which it has a high affinity [38] (Table 2, Figure 2).

An adenosine triphosphate (ATP) proton pump acidifies the endosome after the complex endocytosis, allowing the Fe^3+^ detachment from the holo-transferrin. Fe^3+^ is then reduced to Fe^2+^ by a reductase to enable them to pass through DMT1 [16,39]. Once inside, iron is believed to loosely bind to citrate, ATP, adenosine monophosphate (AMP), or other compounds [40,41]. In these complexes, Fe^2+^ reaches mitochondria where heme and clusters of iron-sulfur are biosynthesized or it is stored in ferritin [23,42,43,44] (Table 2, Figure 2).

All cell types have ferritin stores, but they are most numerous in the bone marrow, liver, and spleen. The stores in the liver, in particular, are the most abundant reservoir of iron. Women use more of their stores since they have to integrate the iron losses due to menstruation and sometimes their stores may be as low as 500 mg. Fe^2+^ excess traveling to the blood is facilitated by transferrin. Fe^2+^ is then oxidized into Fe^3+^ by ceruloplasmin (which acts in the same way hephaestin does in enterocytes) and it is loaded into transferrin. Iron metabolism is complex and strictly regulated by post-transcriptional processes including the IRE/IRP system. The IRE/IRP system is vital since it controls cellular iron uptake as well as its utilization, storage, and transport [45] (Table 2). IREs are an evolutionary conserved mRNA sequence with stem-loop structures of 25–30 nucleotides. During iron deficiency conditions, IRPs bind to target IRE regions, stabilizing the mRNA of TfR1 and steric inhibition of ferritin mRNA translation. TfR1 synthesis promotes cellular iron uptake from circulating transferrin.

Conversely, if ferritin biosynthesis is inhibited, this prevents iron storage and allows metal mobilization. When iron is in excess, IRPs are inactivated, leading to the degradation of the TfR1 mRNA and ferritin mRNA translation. This regulation reduces additional iron internalization via TfR1 and facilitates its loading in cytosolic ferritin [46].

### 2.3. Iron Uptake from the Systemic Circulation to the Brain

Even though crucial for life and brain function, the mechanisms of metal acquisition within the central nervous system (CNS) in humans are still almost elusive. Transferrin transports Fe^3+^ in the circulation, and through the brain’s capillaries. In the capillaries, the brain capillary endothelial cells (BCEC) are tightly joined and constitute the Blood-Brain-Barrier (BBB). On the luminal surface of the BCEC, the complex of transferrin loading iron is taken by the receptor TfR1 (Figure 3A,B), and the transferrin/TfR1 is internalized into an endosome. Inside the endosome, iron is detached from the complex and remains there while the TfR1 travels through the cytoplasm. The endosome then reaches the BCEC’s abluminal side, and it fuses with the cell membrane. This exposes Fe^3+^ that is released to the extracellular interstitial space. The apo-transferrin stays attached to TfR1 and undergoes endocytosis again to travel back to the BCEC luminal side. Here, the apo-transferrin is released into the capillaries, and it is ready to start another cycle of Fe^3+^ ions binding [47,48,49,50,51]. It has been also reported that macrophages are involved in the passage of Fe-proteins between the brain parenchyma and the peripheral circulation, thus playing a crucial role in Fe homeostasis [13].

Iron seems to enter the brain in this way, mostly. However, experiments on hypotransferrinemic mice have demonstrated that iron can travel to the brain also via a Transferrin-independent mechanism [52]. For details on this mechanism, see Mills and colleagues [53].

The BCEC luminal side is in contact with the astrocytes and does not appear to be directly connected with neurons. In the space between the astrocyte and BCEC, Fe^3+^ can bind apo-transferrin or ATP and citrate, as well as other low-weight elements. These complexes are present everywhere in the extracellular space of the brain, but they appear mainly concentrated in the space between the astrocyte and the BCEC side.

Transferrin iron, through the extracellular fluid, can easily reach the area of a neuron, on whose membrane the TfRs are present. Moreover, astrocytic foot processes surrounding the neuron surface express both ferroportin [54] and a Cp-GPI- [55], suggesting an alternative path. Fe^3+^ possibly bound to ATP or citrate can be internalized into the astrocyte. At this point, it could be transported through the astrocyte, reduced by a yet unidentified ferrireductase and transferred outside from the end foot of the astrocyte via ferroportin. Here, ceruloplasmin might catalyze the oxidation of Fe^2+^ into Fe^3+^ (see below), facilitating iron uploading on apo-transferrin. The mechanisms described are not mutually exclusive but can undoubtedly work together to bring iron to the neuron (Figure 3B).

Similarly, to the BCEC, the membrane of the neuron has TfRs that can again initiate endocytosis. The critical variation between the BCEC and the surface of the neuron is that the latter has DMT-1 which becomes engulfed in the endosome and sinks with it [51] (Figure 3B).

Furthermore, ferroportin can also be expressed in presynaptic vesicles [54]. This is in favor of the hypothesis that Fe^2+^ may be released in the inter-synaptic space via presynaptic vesicles. The free Fe^2+^ in the inter-synaptic space may trigger oxidative stress in the interface between the pre-and post-synaptic membranes.

### 2.4. The Key Ceruloplasmin Function as a Copper and Iron Crosstalk Protein

As already mentioned, ceruloplasmin biosynthesis occurs mainly in hepatocytes. The copper pump ATP7B protein transfers six atoms of copper into apo-ceruloplasmin. This allows the ceruloplasmin folding as an active holo-ceruloplasmin that is then delivered into general circulation. Ceruloplasmin is also synthetized endogenously by cells of the CNS and is strongly expressed by astrocytes in the vicinity of neurons [55] in a Cp-GPI form. Ceruloplasmin catalyzes the oxidation of Fe^2+^ to Fe^3+^ via the following vital reaction in our physiology:4 Fe^2+^ + 4 H^+^ + O_2_ → 4 Fe^3+^ + 2 H_2_O(1)

This reaction has the effect to reduce the bioavailability of Fe^2+^. The metal is bussed to various tissues and organs by transferrin, which loads only Fe^3+^ [56,57]. Mostly, ceruloplasmin detoxifies potentially toxic iron since the by-product of equation 1 is simply water.

Ceruloplasmin has another essential role that is to scavenge H_2_O_2_ [58], thus preventing the generation of oxygen radicals. This is because non-enzymatic oxidation of Fe^2+^—that could take place at physiological pH—could trigger dangerous Fenton reactions. Ceruloplasmin can then also prevent lipid peroxidation.

The amount of copper available in the liver regulates the synthesis of holo-ceruloplasmin. The lower the copper, the lower the ceruloplasmin. [59]. Furthermore, reduction in copper bioavailability produces a certain amount of apo-ceruloplasmin, which has no ferroxidase activity and is quickly degraded in plasma. Mutations of the *ATP7B* gene result in low holo-ceruloplasmin levels, as exemplified in Wilson’s disease, which is the paradigmatic disease of non-ceruloplasmin copper toxicosis or accumulation in both the liver and brain [60,61]. In addition, functional single nucleotide polymorphisms in the *ATP7B* gene can result in alteration of ceruloplasmin activity or specific activity reduction mainly associated with AD. Instead, a mutation in the *CP* gene, encoding for ceruloplasmin results in aceruloplasminemia (Reviewed in [62]).

Decreased levels of ceruloplasmin determine a reduction of ferroxidase activity that results in more free Fe^2+^ available to catalyze Fenton reactions. For this reason, aceruloplasminemia results in severe iron overload. Apoceruloplasmin is thought to be a disorder of iron homeostasis. It shares symptoms with hemochromatosis. To avoid misdiagnosis, the measurement of transferrin saturation may be of help as transferrin saturation is low in aceruloplasminemia and high in hemochromatosis. A complication of aceruloplasminemia is an increased lipid peroxidation and an impaired fatty acid oxidation.

Ceruloplasmin is a determinant for establishing the iron oxidation rate of serum. Oxidation is needed for bussing iron from tissue storage to general circulation [63,64,65]. Ceruloplasmin thus represents a crosstalk protein between iron and copper metabolism. Experiments made in the sixties on perfused liver preparations showed that the addition of ceruloplasmin markedly facilitates iron release in blood circulation. This suggests that ceruloplasmin is critical for iron mobilization [65]. Thus, ceruloplasmin is a complex protein playing such an important role in antioxidant defense (Figure 4), and consequently, involved in the molecular mechanisms of liver diseases and neurodegenerative disorders [66,67].

### 2.5. When and How Does Iron Become Toxic?

As we have seen so far, the body undergoes a sophisticated sequence of diverse processes to strictly regulate the absorption of iron since any deregulation of the physiological balance can quickly generate free radicals, which in turn cause disruption of essential protein processes, tissue damage and abnormal accumulations, generating by the overall oxidative stress [68,69]. Free radicals are molecules with unpaired valence electrons, making them highly reactive [70]. Oxygen produces some of the most active radicals, i.e., ROS. Oxidative stress is defined as the harmful impact on cellular function or viability as a consequence of the loss of balance between ROS and antioxidants in the cellular environment. An excess of ROS can induce damage to biomolecules, including proteins, DNA, and the nucleotide pool. The 8-oxo-7,8-dihydro-2′-deoxyguanosine (oxo8dG) is the prevalent oxidized form of the base modification produced by the reaction with the hydroxyl radical at the C8 position of the 2′-deoxyguanosine (dG) incorporated into DNA or Guanine incorporated into the nucleotide pool, in triphosphate forms (dGTP or GTP). Among the most important ROS is the superoxide anion O_2_^−^ produced naturally by metabolic processes or artificially by irradiation. O_2_^−^ is important since it interacts with other molecules via enzyme- or metal-catalyzed processes to generate ROS further. For example, our immune system produces superoxide anions and nitric oxide in response to inflammation processes. The two easily react with each other to produce the peroxynitrite anion which is known to cause lipid oxidation and DNA fragmentation.
NO^•^ + O_2_^−^ → ONOO^−^(2)

The superoxide anion also reduces Fe^3+^ to Fe^2+^ by via the reaction:Fe^3+^ + O_2_^−^ → Fe^2+^ + O_2_(3)
where Fe^2+^ is highly dangerous since it can quickly enter a Fenton reaction, which produces the hydroxyl radical ^•^OH, one of the most reactive of all ROS species.
Fe^2+^ + H_2_O_2_ → Fe^3+^ + ^•^OH + OH^−^(4)

Moreover, the above reaction easily proceeds as follows:Fe^3+^ + H_2_O_2_ → Fe^2+^ + ^•^OOH + H^+^(5)
which can again proceed as in Equation (3). In other words, the encounter of Fe^2+^ with peroxide can trigger a cyclical production of vicious ROS species. This is the reason why our body effectively tries to minimize the circulation of iron in its Fe^2+^ form, as can be gathered from the mechanisms of absorption and distribution described in the preceding paragraphs.

Noteworthy also copper is involved in Fenton reaction in an identical manner:Cu^+^ + H_2_O_2_ → Cu^2+^ + ^•^OH· + OH^−^
(6)

One of the reasons why radicals such as ^•^OH are vicious is that they promote peroxidation of polyunsaturated fatty acids: the initiation phase removes hydrogen from the unsaturated lipid and produces a lipid radical R(^•^)
R(H) + ^•^OH → R(^•^) + H_2_O(7)

This propagates as
R(^•^) + O_2_ → R(OO^•^)(8)
which reacts with another unsaturated lipid, terminating the fatty acid peroxidation and by producing an additional lipid radical:R(OO^•^) + R(H) → R(OOH) + R(^•^)(9)

However, the lipid radical R(^•^) can engage a chain peroxidation at this point by participating again in the reaction (8).

### 2.6. Metals in Neurological Disorders

In the last 20 years, the number of neurologic studies on metals has increased extraordinarily. Transition metals have shown essential functions in neurobiology because they alter the redox state of the cellular environment, catalyze redox reactions [71] biologically, and disrupt neuronal architecture [69]. In particular, iron, copper, and also zinc contribute to diverse neurologic processes, including energy metabolism (iron, copper); antioxidant defense (copper, zinc); myelination (iron); DNA synthesis (iron, zinc); neuronal cytoskeleton integrity (zinc); and neurotransmitter synthesis (iron, copper, and zinc) [72]. Genetic studies of inherited metal metabolism disorders have provided insights into some metal-related altered pathways, resulting in brain dysfunction, helping to understand the role of metal homeostasis in brain functionality [64,73].

Inherited diseases related to iron metabolism are more common and more numerous than those related to copper and some of them show neurologic phenotypes. Aceruloplasminemia results in progressive dementia, dysarthria, and dystonia which are secondary to basal ganglia iron accumulation [74]. Studies on patients with *HFE* hereditary hemochromatosis highlighted the presence of iron accumulation in the cerebral cortex, hypothalamus, lentiform nucleus, dentate nucleus, basal ganglia, substantia nigra and in the red nucleus, suggesting that *HFE* mutations modify the risk of developing neurodegenerative disorders [75]. Patients with neuroferritinopathy show basal ganglia nuclei with cavitary degeneration, neuronal loss, and microglia with the inclusion of iron and ferritin [64]. Genetic variants of the *HFE* gene, notably, the H63D variant have been reported to act as a potential risk factor for AD patients [76,77]. Lehmann et al. revealed that *HFE* 282Y in combination with TF C2, and possibly also *HFE* 63HH with TF–2AA might lead to iron overload and eventually oxidative stress generation in the preclinical phase of AD [78]. It has been proposed that *APOE4* and age influence the former and the latter combination, respectively. Notably, at the cellular level, the HFE mutant protein formed from the *HFE* H63D gene variant has been found to link with iron dyshomeostasis, enhanced oxidative stress, glutamate release, tau phosphorylation, and variation in inflammatory response, which has been considered as contributing factors for the onset of neurodegenerative diseases.

Friedreich’s ataxia is associated with sensory neuron and cerebellar degeneration due to the effect of impaired mitochondrial iron homeostasis on neuronal survival [79]. Mutations in gene encoding for other regulatory proteins (i.e., *PANK2*, *PLA2G6*, *FA2H*, *ATP13A2*, *DCAF17*) have been correlated with basal ganglia iron accumulation which results in different neurologic symptoms (e.g., dystonia, dysarthria, pigmentary retinopathy) [64].

When neurodegenerative diseases, e.g., AD, Parkinson’s disease (PD), and multiple sclerosis (MS) are studied in relation with these inherited metal-metabolic disorders and, specifically, to their main neurological traits, it seems that several of their typical signs resemble those of the inherited mental disorders. In other words, neurodegenerative diseases have a complex etiology and share enough traits with inherited metal-metabolic disorders, to assume genetic similarities between them (Table 3).

The role of metal biology and iron accumulation has been claimed to be involved in PD as well [80]. *CP* gene variants have been related to PD [81]. Specifically, some authors demonstrated that increased iron deposits can be seen in the brain’s substantia nigra by transcranial ultrasonography (hyperechogenicity). These deposits were associated with *CP* gene variants [81]. However, studies on humans indicated a high heterogeneity among the results on systemic variations in metals between PD patients and healthy controls.

The involvement of metal disarrangements in neurodegenerative disorders has made them an emerging target for studies on early diagnosis [75,82,83], environmental exposures [84,85,86], prevention strategy, and therapeutic interventions [87].

### 2.7. AD and the Role of Iron

Hepcidin, Ferritin, and IL-6 participate significantly in host defense mechanisms associated with neuroinflammation in AD. Hepcidin was found elevated in AD serum patients, indicating its correlation with the disease [13]. Several studies reported metal homeostasis abnormalities in AD patients’ brain, blood, and CSF. In a cohort of 94 participants, ferritin, both in plasma and CSF was shown to have the potential to discriminate AD in the preclinical phase, classified into low and high neocortical Aβ load groups, prior to cognitive impairment [88]. The rise in ferritin levels suggests increasing iron levels in CSF and brain that can be associated with ferroptosis, an iron-dependent cell death that results from a build-up of lipid peroxides and is regulated by Glutathione peroxidase 4 (reviewed in [89]). In addition, transferrin levels have been revealed to increase the risk of AD, decreasing the protein associated with a 12% risk of AD [90].

In addition, ceruloplasmin has been shown to predict cognitive decline and brain atrophy in people with underlying Aβ pathology [91] when evaluated in the CSF or when measuring ceruloplasmin specific activity in serum [90]. Consistent with these findings, our previous results showed that AD patients with a severe burden of medial temporal atrophy had increased systemic concentrations of iron and an increased percentage of TSAT, at a difference from frontotemporal lobar degeneration patients [92]. Furthermore, AD patients exhibited the activation of the ceruloplasmin-transferrin (Cp-Tf) system [93]. Studies in the early 1980s claimed the Cp-Tf system as the primary antioxidant system acting in plasma [94]. Some authors [94] clearly demonstrated that the Cp/Tf ratio can reflect the combined antioxidant activity of ceruloplasmin bonding a copper in the oxidized state (Cu2^+^) [95] and of the apotransferrin [96], both measured as Electron Paramagnetic Resonance (EPR) spectrometry. The Cp/Tf ratio resulting from the measure of concentration units can reasonably represent the Cp-Tf system’s functionality, which is otherwise expensive to be monitored by EPR spectrometry [94]. Some studies in experimental hypercholesterolemia showed that the activation of the Cp-Tf system can reduce lipid peroxidation [94]. The activation of the Cp-Tf system suggests an increased oxidative stress in AD patients that afflicts patients as a systemic condition [93]. Consistent with these results, ceruloplasmin, peroxides, and Cp/Tf were elevated in AD patients and correlated inversely with Mini-Mental State Examination (MMSE) scores [93,97], while medial temporal atrophy correlated negatively with serum levels of iron and positively with Cp/Tf [93]. AD patients have also been reported to exhibit lower albumin, longer prothrombin time, and higher transaminases ratio (Aspartate/Alanine transaminases, AST/ALT) values than healthy controls. This indicates a liver hypofunction, along with a reduction of transferrin and an increase in serum ferritin levels [82]. AD carriers of *H63D* mutation in the *HFE* exhibited increased levels of iron and decreased levels of transferrin and ceruloplasmin, resembling hemochromatosis, which was not found in *H63D* non-carrier AD patients, suggesting that carrying the *H63D* mutation seems not to be itself sufficient to increase the risk of AD. Rather, it is a synergy between more factors together: a condition of liver dysfunction, *H63D* genetic mutation, and an iron increase that might boost the risk of AD [82]. These findings are consistent with the recent formulation of the hypothesis of iron involvement in AD, depicting abnormalities of iron homeostasis as a central driver of the AD risk [98], as it will be discussed in Section 2.9.

### 2.8. Neuroimaging Studies of Iron in AD

As discussed in detail, iron stored in ferritin or hemosiderin in the brain is highly indispensable for critical physiological processes; however, iron overload can promote the free radical generation and oxidative damage. Consistently, it has been demonstrated that an age-related increase in stored iron can promote common medical conditions such as diabetes and vascular disease which are associated with an increased probability for AD development by magnifying the redox-active iron pool in brain cells. The increase of iron uptake in the AD brain may be due to the generation of a more stable IRE/IRP1 complex which is expected to enhance the stability of TfR1 mRNA [99]. Moreover, intracellular iron storage in ferritin may be altered in the AD brain [80,99,100,101].

Increased stored iron due to high iron load and impaired iron storage/detoxification could aggravate Fenton reaction-mediated oxidative stress/free radical damage in vulnerable neurons, a critical sign of early change in AD. Moreover, iron-induced oxidative stress and impaired cell signaling could favor phosphorylation of microtubule-associated protein tau. Neuritic plaques and neurofibrillary tangles are considered the main histological features of AD. At the same time, elevated iron levels have been linked with an increased formation of Aβ peptides which are the basic components of neuritic plaques. Thus, elevated levels of brain iron have been correlated with an increased risk of AD; however, it is not clear whether iron deposition in the brain is a major cause of AD. Consistent with this fact, several studies have tried to monitor the spatial distribution and alterations in iron levels to understand the pathogenesis of AD. Previous studies have observed an association of iron with plaque in post-mortem studies using magnetic resonance imaging (MRI) transverse relaxation of brain slices [102] and histochemistry [103].

A similar pattern of results is also found in the brains of APP/PS1 mice using X-ray fluorescence [104]. Ayton et al. have demonstrated that high brain iron is associated with longitudinal cognitive decline and Aβ burden in living AD patients using mixed-modality MRI and PET imaging studies [105]. The same research group has recently shown that CSF ferritin level is longitudinally connected with worse cognitive performance and might facilitate Aβ deposition in subjects with biomarker-determined AD pathology [106]. It has been proposed that ferritin levels might be influenced by apolipoprotein E4 [107], which is suggestive of an increased interactions between Aβ and iron [108], responsible for oxidative stress and Aβ aggregation [109,110]. Moon et al. and Du et al. have also reported an increased brain iron deposition in the putamen and caudate nucleus of AD patients using MRI and quantitative susceptibility mapping (QSM), showing specific interest in association with AD [102,111].

The possible reason may be related to iron accumulation in the brain that is closely correlated to the neurofibrillary tangles and Aβ fibrils. In addition, endogenous iron and zinc may facilitate the aggregation of Aβ fibrils and/or an existence of iron-responsive elements in the mRNA encoding APP. An in vivo study has demonstrated that iron deposits with different morphology and magnetic/oxidation states seem to be integral to the plaque-like regions. Using an integrative set of advanced electron analytical microscopy, mineralized iron was present as iron oxide (Fe_3_O_4_) magnetite nanoparticles within amyloid plaque cores (APC) from post-mortem cases of AD, providing evidence of association between iron accumulation and Aβ aggregation in AD [112].

Consistently, Telling and colleagues have revealed the presence of high amounts of reduced ferrous (Fe^2+^) iron and magnetite deposits co-localized with amyloid structures in the cortex of the APP/PS1 mouse model of AD by applying advanced X-ray microscopy techniques [113]. Notably, there were very few iron deposits in cortical sections of wild type mice. The enrichment of Fe^2+^ may act as a potential source of free radical generation including highly reactive hydroxyl radical by Fenton reaction. It is believed that Aβ in the plaque may induce redox cycling Fe^3+^ into a pure Fe^2+^ and thereby dynamically participate in the mineral deposition [114]. Moreover, the deposition of iron can potentially favor the aggregation of Aβ which has been already shown by previous in vitro [114] and in vivo studies [115,116]. A pilot study conducted by Sternberg and colleagues revealed that iron and iron-related protein levels were high in the upper 50% as compared to controls after patient stratification based on clinical dementia, indicating that iron dyshomeostasis could aggravate cognitive impairment. Moreover, pure AD subjects display three times higher serum hepcidin levels than controls. Hence, both hepcidin and iron-related proteins are considered as a group of serum biomarkers that relate to diagnosis and progression of AD [117]. A recent study has observed a heterogeneous magnetite/maghemite distribution pattern across AD and cognitively normal brains from northern England using superconducting quantum interference device (SQUID) magnetometry [118].

### 2.9. Genetics of Iron in AD Pathogenesis

Biometals, such as iron, copper, and zinc, are finely regulated in the brain. Their neurotoxic effects are not merely due to increased exposure but rather to disarrangement in their homeostasis and related compartmentalization in oxidative stress or processes of excitotoxicity. Deregulation of brain metal metabolism is supposed to be caused by non-genetic as well as genetic factors. It may also arise at different levels (i.e., uptake and release, storage, intracellular metabolism, and regulation) [119].

The increased knowledge about the human genetic variation allowed us to discover several Loss-of-Function (LoF) variants in genes encoding for regulatory proteins of metal metabolism, permitting us to validate the interest in metallobiology of neurodegenerative diseases as initially described by biochemical criteria [120]. In fact, some neurological disorders with a complex disease etiology seem to share traits that typically distinguish inherited metal disorders, as described previously. In this paragraph, we will discuss this matter, specifically addressing the topic of metal genes, their inherited disorders, and AD. Both overload and deficiency have been claimed to cause neuronal dysfunction concerning iron metabolism. In the brain, the proteins mostly involved in iron homeostasis are: HFE, ferritin, transferrin, TfR, IRP, DMT1, and ceruloplasmin [75]. By using the power of genetics, a number of studies have explored the role of these iron-related LoF variants in neurodegenerative disorders and specifically in AD [75,121]. One of the most investigated hypotheses concerns mutations in the HFE gene, which accounts for the iron increased deposits reported in AD. HFE is a protein that regulates the absorption of iron. Mutations in the *HFE* gene cause hemochromatosis typified by an increased absorption of iron from the diet and accumulation in tissues and organs, including brain [31]. Although HFE protein is expressed in different brain regions and influences brain iron uptake, the relationship between HFE mutations and neurodegenerative disorders did not receive attention until 2000. According to this hypothesis, the first report on *HFE* mutations and AD risk was published [122]. The authors showed that frequencies of the *HFE* mutation were increased in men affected by familial AD and among non-carriers of the *APOE*4 allele with respect to healthy aged men. Then, a number of studies followed that explored the role of *HFE* mutations in AD risk, reporting significant associations with AD features, for example, disease onset [123,124], cognitive symptoms [125], the severity of clinical deficits [126], markers in the CSF [76], and conversion from mild cognitive impairment (MCI) to AD [125]. However, the diverse studies did not reach univocal results depicting a conflicting picture likely related to inter-ethnic differences in *HFE* allele frequencies and the interaction with genetic, environmental, and demographic factors that may confound the genetic association [75].

Additional studies that investigated other iron-related genes (*TFR*, *DMT1,* and *IREB2*), found positive associations with some of them with AD phenotypes [78,127]. Moreover, a common protein variant of HFE, H63D HFE, has been recognized as a modifier of multiple neurological disorders [128]. Nevertheless, as for *HFE* gene also for these regulatory genes, further studies are needed to confirm the association and determine if the gene variants analyzed have a functional effect on iron storage and deposition. Finally, studies in mouse models have demonstrated that defects in *FTH1* (encoding for ferritin), *CP,* and *TFR* genes were associated with the typical signs of neurological disorder phenotypes [129]. Even though these outcomes highlight that LoF in iron related genes may have effects on the pathogenesis of AD, additional genetic association studies on humans are necessary to confirm their role.

Despite the fact that several studies have analyzed copper concentration in diverse human matrices—for example, serum, plasma, CSF, brain, liver, concerning neurodegenerative diseases and AD specifically [130]—information about the direct role of copper related genes in AD is still scanty, and limited to the *ATP7B* gene [131].

To date, no Genome-Wide Association Studies (GWAS) on AD have found a significant variant in genes related to metal regulation [132] or metallome-dependent pathways [133] (www.alzgene.org, accessed on 29 August 2022). This situation can be due to the complexity of metal homeostasis regulation, in which genetic and environmental components strongly interact with each other. Therefore, a candidate gene approach coupled with biochemical and environmental investigations may help understand the role of metal regulatory genes in AD pathogenesis. Supporting this, a direct connection between iron homeostasis and regulation of APP processing has been revealed. APP has an IRE in the 5′-untranslated region of the APP transcript that promotes the translation of APP in response to iron [134].

### 2.10. Ferroptosis a New Neurodegenerative Process Involved in AD

With a deep focus on iron involvement in AD, a recent longitudinal study [98] evaluating iron in AD brains confirmed that tissue iron load could affect NFT formation, at least in the inferior temporal cortex, even if the results did not show an association between iron load in bulk tissues and Aβ plaques. Three brain areas were primarily affected in AD: the anterior cingulate cortex, the mid-frontal cortex, and the inferior temporal cortex, as well as the comparatively spared cerebellar cortex. A total of 645 post mortem AD brains were investigated to test for the association between brain iron and pathological and ante mortem clinical changes in AD [98]. The study revealed that the burden of iron in the brain was modestly associated with the clinical diagnosis and neuropathology of AD but showed a strict association with the rate of cognitive decline in the decade that preceded patient’s death. The results argue for an additional downstream role for iron as an effector of neurodegeneration, independently of tau or amyloid pathologies.

These data inspired a new formulation of the *Metal Hypothesis of AD* [135], which posited that iron promotes plaque and tangle formation by promoting APP and tau production as well as aggregation of Aβ and P-tau. The new hypothesis posits that iron has an additional role downstream of proteinopathy, by influencing the susceptibility of neurons to die due to ferroptosis. Ferroptosis is a unique identified non-apoptotic cell death, characterized by glutathione depletion and lipid peroxidation. The occurrence of ferroptosis is not related to the increase of iron to toxic levels but, instead, iron becomes activated during ferroptosis by lipid peroxidation to induce toxicity. Iron, therefore, acts as a moderator of susceptibility, and the cell death occurs once ferroptosis is activated. Recent studies have shown that ferroptosis is closely associated with pathophysiological processes of many diseases, such as AD, PD, and amyotrophic lateral sclerosis (ALS) [136,137].

Until now, research on the topic of iron in AD has investigated the metals’ role in promoting hallmark pathology. The discovery of ferroptosis provided a new explanation for the neurodegeneration in AD, which occurs after the appearance of the proteinopathy.

Currently, reports have started emerging for the role of ferroptosis in AD [138]. If confirmed by further clinical evidence, this hypothesis could have significant implications not only for the understanding of the pathophysiology of iron in AD but above all for the possibility of providing new therapeutic opportunities that could target iron pathways [139,140].

Ferroportin1 (Fpn), the only mammalian non-heme iron exporter identified, is down regulated in the brains of APPswe/PS1dE9 mice as a mouse model of Alzheimer’s and AD patients. Genetic deletion of Fpn in neurons of the neocortex and hippocampus by breeding Fpn (fl/fl) mice with NEX-Cre mice led to phenotypic characteristics of AD as the hippocampal atrophy and presence of memory deficits. Ferroptosis was observed in both Fpn (fl/fl/NEXcre) and AD mice. Gene set enrichment analysis (GSEA) of ferroptosis-related RNA-seq data showed that the differentially expressed genes in ferroptosis were highly enriched in AD-related gene sets. The main functions of the enrichment genes were distributed in several important pathways relative to the function as ROS metabolic processes and amyloid beta formation, supporting the important role of ferroptosis in AD pathogenesis [138]. New evidence suggests that iron chelators can suppress ferroptosis [140] or prevent it by inducing a lipid repair system involving glutathione and glutathione peroxidase 4 that converts lipid hydroperoxides to lipid alcohols [141].

### 2.11. Fe and APP/Aβ System Metabolism

It is well known that the AD brain’s hallmark is the presence of Aβ plaques and neurofibrillary tangles, respectively, outside and inside the neuronal space. As mentioned in the Introduction, the amyloidal aggregations’ major constituent is Aβ, a 39–43 amino-acid peptide, produced from the cleavage of the APP. More specifically, it has been reported that APP can be processed into two separate pathways: the non-amyloidogenic and the amyloidogenic. In the non-amyloidogenic pathway, APP is cut at two sites placed in the extracellular space and within the transmembrane domain, α-secretase and then by γ-secretase, respectively, releasing a truncated P3 fragment. In the amyloidogenic pathway, APP is first cleaved by β-secretase at a site placed outside the cell surface, generating a C-terminal fragment that is subsequently cleaved by γ-secretase at a site placed within the transmembrane domain. The product of this cleavage is Aβ [reviewed in 8] (Figure 5). APP is virtually ubiquitous and has been reported to have a role in iron export by binding ferroportin on the plasma membrane [142].

Furthermore, the mRNA APP transcript has an IRE sequence in the 5′-UTR. When cellular iron conditions are low, IRP 1 binds the IRE of APP mRNA. APP mRNA is then not translated into APP protein, thus down regulating APP-mediated stabilization of ferroportin that further inhibits iron efflux [143]. Additional evidence in this regard, comes from the preclinical model of AD showing that *APP* knockout mice display accumulation of brain iron [144], indicating that in the healthy brain, APP may take part in iron homeostasis.

The current hypothesis in the literature is that iron promotes plaque and tangle formation by promoting APP and tau production as well as Aβ and phospho-tau aggregation. Accordingly, new evidence from 645 post mortem brains confirmed that tissue iron concentration might impact neurofibrillary tangles formation, at least in the inferior temporal cortex. This investigation of bulk tissue iron levels did not support a relationship between bulk tissue iron concentration and amyloid formation. Moreover, it is suggested that the loss of tau function could cause neuronal iron accumulation [145].

When it became clear that a significant contribution to pathological amyloidal deposition comes from oxidative stress—generated by excessive ROS activity [146] and indirectly by the products of lipid peroxidation [147]—the attention of AD researchers started to be focused on transition metals; they take part in a variety of chemical reactions that generate uncontrollable ROS, capable of damaging and destroying molecular and cellular compartments [9]. It is now established that APP is a copper protein with copper and zinc-binding domains, which mediate redox activity, also involved in Aβ aggregates formation [148,149]. Aβ too has selective copper and zinc domains that bind equimolar quantities of the two metals in normal conditions. However, Aβ zinc is completely displaced by copper during acidosis [150].

According to several studies that have demonstrated increased concentrations of iron related to the presence of Aβ plaques: (1) Enhanced concentrations of both iron and copper have been reported within Aβ plaques and neurofibrillary tangles [151,152]; (2) Deposits or increased concentrations of iron have been found in the basal ganglia [153,154], in the CSF [155], and in other brain areas affected in AD [98]; and (3) The hippocampus chiefly and the brain cortex secondarily, are affected by Aβ-metal toxicity.

By now, this view has become accepted as a possible explanation for the toxic properties gained by Aβ, supporting a general agreement on the existence of a link between AD and oxidative stress phenomena, triggered by transition metals as a possible pathway of AD pathogenesis (Metal Hypothesis of AD [136]).

## 3. Current Perspectives of the Therapeutic Options Targeting Iron Chelation in AD

Mounting evidence has reported that about half of all proteins are needed to bind with metals and form metalloproteins for effective functions. As previously discussed, metal ions in the brain are required in the appropriate functioning of enzymes, neurotransmission, and aging [72]. On the other hand, there are also accumulating studies which link biometal (iron, copper, zinc) dyshomeostasis and metal-amyloid interactions to various neurodegenerative diseases including AD [112]. Accordingly, AD is suggested to be an ailment of protein aggregation. Yet, it is also considered as a disease of metal dyshomeostasis. There is a common consequence of the binding of copper, iron, and zinc to APP. Copper (and iron) can bind to the amyloid protein and enhance the risk of oxidative damage by generating ROS which directly overcomes the antioxidant defense system in the neurons [156]. Additionally, zinc is known to bind both APP and Aβ. The binding of zinc to APP occurs at the Lys16 which may affect the cleavage activity of α-secretase and thus reduce the assembly of soluble APPα and enhance Aβ production [157]. However, brain zinc levels are very strictly controlled and the major consequence of zinc therapy overdosage (zinc toxicity) or long-term consumption of excessive zinc is copper deficiency. On the other hand, raised levels of iron in the neuropils of an AD brain are intensely associated to pathology via the generation of ROS [156].

However, since the role of biometals in the development and progression of AD is very complex, it is still controversial whether AD is associated with excess or deficiency of iron. Both less and excess levels of metals have been linked with neurological disorders including AD [158,159]. Excess levels of biometals such as copper, iron, and zinc accumulate in amyloid plaques of AD patients with up to 5.7, 2.9 and 2.8 folds, respectively, in comparison to normal brains [160]. Conversely, Brewer et al., have observed lower serum levels of zinc in AD patients as compared to the control group [158]. There is also evidence for the claim of iron dyshomeostasis in AD, including alterations in iron, ferritin, and transferrin [89,90]. Low brain iron levels are associated with motor neuron deficits, altered dopamine activity, and aberrations in myelinogenesis [161]. As evident from the literature, iron can bind with hyperphosphorylated tau and thereby facilitate its aggregation and lead to the formation of NFTs [145]. In contrast, several in vitro studies found that iron decreases Aβ aggregation and neuritic plaque formation [162]. However, excess levels of iron pose a threat to the brain especially due to its redox activity. As already discussed, iron participates in Fenton reactions and promotes the generation of hydroxyl and superoxide radicals that interact with proteins, lipids, and DNA [68,69]. Hence, extremely evolutionarily conserved mechanisms are required to preserve brain iron homeostasis.

Since the role of biometal ions and their interaction with amyloid are critical to the pathogenesis of AD, the assessment of high-affinity metal-neuronal protein interactions may favor identifying novel therapeutic approaches for the treatment of AD and other neurodegenerative diseases. Recently, chelating agents have been reported to dislodge interactions between metal ions and proteins which thereby reduce oxidative stress and may improve cognition [163]. Some recent evidence revealed that chelators of copper, iron, and zinc effectively inhibit Aβ fibrils aggregation [163,164,165,166]. Certainly, metal chelation treatment deals with defense against acute as well as occasionally chronic oxidative damage and impedes protein aggregation. Fasae and his colleagues have discussed the clinical use of metal (copper, iron, zinc) chelators in AD [166]. Deferoxamine (DFO) is a trihydroxamic acid, a chelating agent that displays a high affinity for Fe(III) in comparison to other metals and is considered a precise chelator for iron overload ailments such as thalassemia. Deferiprone (L1) (1,2-dimethyl-3-hydroxypyrid-4-one) and Deferasirox (4-[3,5-bis(2- hydroxyphenyl)-1,2,4-triazol-1-yl]-benzoic acid (Exjade, ICL-670) are offered alternatives to DFO with high plasma life, more gut tolerance, and patient compliance [167] which treat transfusion iron overload effectively as compared to DFO [168]. According to a previous study, a remarkable improvement of AD patients has been observed after receiving DFO (125 mg i.m. 2× daily/5 times/week for 2 years) in comparison to the placebo group [169]. However, there is no recent data available regarding competent iron chelators in AD. Various studies have revealed the pharmacology and therapeutic potentials of oral iron chelators (Deferoxamine, Deferiprone and Deferasirox) in both human and rat models [170,171]. EDTA (ethylenediaminetetraacetic acid), a polydentate chelator has a high affinity for divalent and trivalent metal ions such as calcium, magnesium, iron, zinc, and manganese. Klang and colleagues have recently observed that administration of calcium disodium EDTA (CaNa_2_EDTA) reversed the protein aggregation and neurotoxicity caused by iron, copper, and zinc in *C. elegans* and also improved the lifespan of the nematodes [172]. Moreover, the combination of DFO, EDTA and DPA showed an improvement of survival and locomotor activity in flies exposed to iron, copper, and manganese [173]. Multifunctional compounds (MCs) are classified based on their chelating group moieties as bidentate chelators or tridentate chelators [174]. For example, tris (DOP) derivative L1h6 (benzene-1,3,5-tricarboxylic acid tris-2-(3, 4-dihydroxyphenyl)-ethyl]-amide) possess promising geometric arrangement on coordination to Fe^3+^ ion [175]. Tris (DOP) derivative compounds display antioxidant properties and thereby retard oxidative stress-mediated toxicity [176]. The chemically modified lipophilic ferrichrome and tris-hydroxamate chelators are reported to bind iron and decrease iron overload 25-fold compared to DFO [177,178]. Apart from these natural compounds such as flavonoids, natural antioxidants are known to possess iron and copper-chelating abilities. Quercetin [179] and curcumin [180] have been observed to be structurally advantageous to the chelation process as well as in the treatment of animal models and AD patients [181]. Other compounds such as native neuronal peptides such as neuroprotective peptide (NAP) bind to hydroxamate or 8-hydroxyquinoline moieties and forms complexes with Fe^2+^, Cu^2+^ and Zn^2+^ at physiological pH in the water [182]. Furthermore, nanoparticles were also discovered for their capability to cross the BBB via receptor-mediated systems and their achievement in drug delivery [183]. Chelator– nanoparticles were also seen to carry chelator– iron complexes out of the brain reducing the iron ions’ toxicity [184]. There is also evidence that nanoparticle-chelator complexes have been shown to sequester iron or its storage protein such as ferritin from AD brain tissues [185].

Most chelation agents also display several mild or serious side effects and limitations by holding/displacing functional essential metal ions, altering metalloenzyme function, resulting in a poor clinical recovery as well as pro-oxidant activity. However, with regard to another metal, copper, which is also strongly associated with AD, the use of zinc therapy might have beneficial effects. As discussed extensively in [186], zinc therapy acts as a copper competitor and decreases copper concentrations in the body by stimulating the expression of metallothionein, thus preventing direct chelation of the metal. With regard to Wilson’s disease, zinc therapy could have positive effects in counteracting cognitive decline without the occurrence of serious adverse events caused by chelation therapy. A clinical trial (ZINCAID, EudraCT 2019-000604-15) on mild cognitive impairment is still ongoing.

## 4. Conclusions

From the above-described work, we can conclude that an iron disarrangement occurs in AD. Iron regulation is linked to the amyloid formation and tau proteinopathies of AD. The metal can be a causal factor in the onset of disease by promoting ferroptosis or oxidative stress via Fenton reactions, and in association with Tau, APP, and APOE metabolism that have been implicated in physiological iron homeostasis. The findings that iron levels within the normal range are associated with an increased risk of AD—and with disease progression when the proteinopathy is already present—suggest that the metal might also be involved in downstream processes of neurodegeneration. Additional research is warranted to explore these issues in the attempt to set up new therapeutic interventions to contrast AD onset and progression.

## Figures and Tables

**Figure 1 biomolecules-12-01248-f001:**
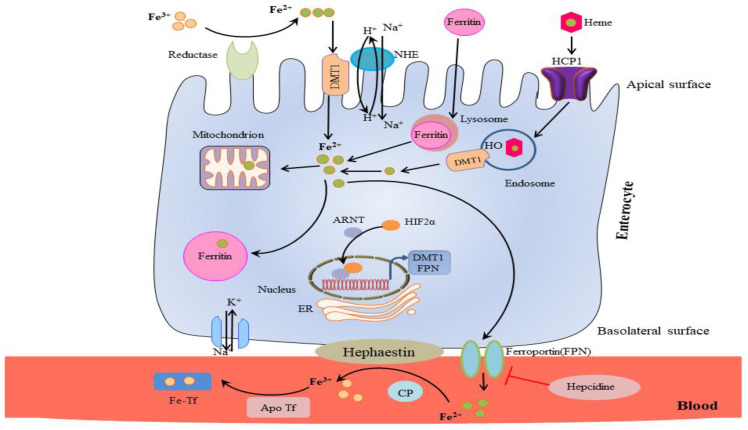
Iron absorption through the enterocyte. Reductases reduced Fe^3+^ before entering in the enterocyte. DMT1 (divalent iron transporter) is a Fe^2+^ transporter that exploits an electrochemical gradient (not shown). The divalent iron can also be absorbed as part of the heme through the heme carrier protein 1 (HCP1) which is also present on the apical surface of the enterocyte. A heme-oxygenase cleaves heme and allows the release of Fe^2+^, then stored in ferritin. Ferroportin facilitates iron transfer to the portal plasma on the basolateral enterocyte surface. The ferroxidase hephaestin (HP) then re-oxidizes Fe^2+^ into Fe^3+^ which binds to apotransferrin (apo Tf) forming holotransferrin (Tf), capable of transporting iron in the blood. Excess iron is exported from the intestine by hepcidin through ferroportin internalization and degradation.

**Figure 2 biomolecules-12-01248-f002:**
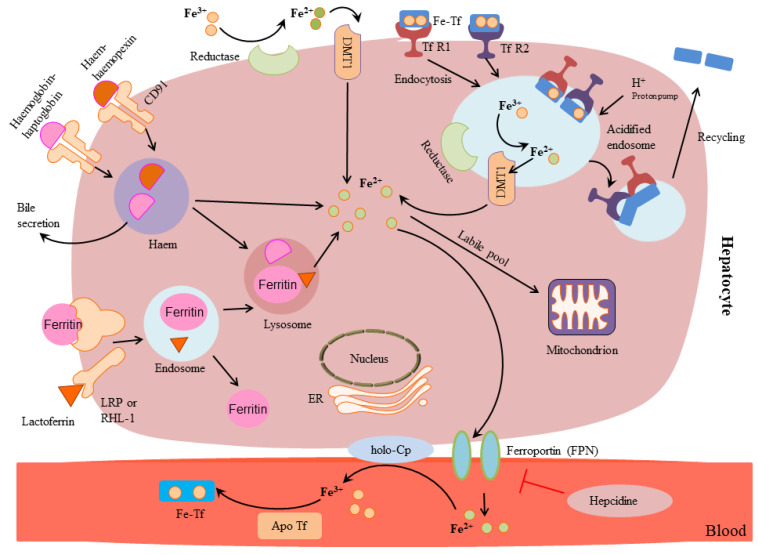
Iron metabolism in the hepatocyte. Iron is taken up by hepatocytes through an endocytosis mediated by the transferrin receptor1 (TfR1): TfR1 is located on the cell surface and binds to a holotransferrin (Fe-TfA) proton ATPase pump acidifies the endosome after the endocytosis, allowing Fe^3+^ release from the holo-Tf. Fe^3+^ are then reduced to Fe^2+^ by a reductase: Fe^2+^ can then pass through DMT-1. Iron is thought to form labile bonds with citrate adenosine triphosphate (ATP), adenosine monophosphate (AMP), or other peptides (labile pool) once inside into the endosome, and to reach the mitochondria. In the mitochondria, iron is used for the biosynthesis of heme and iron-sulfur clusters. It can be also stored in ferritin (the main source of reserve iron). Ferroportin accompanies excess iron, as Fe^2+^, into the blood, where ceruloplasmin (holo-Cp) oxidizes it to Fe^3+^ facilitating the loading onto transferrin (Tf).

**Figure 3 biomolecules-12-01248-f003:**
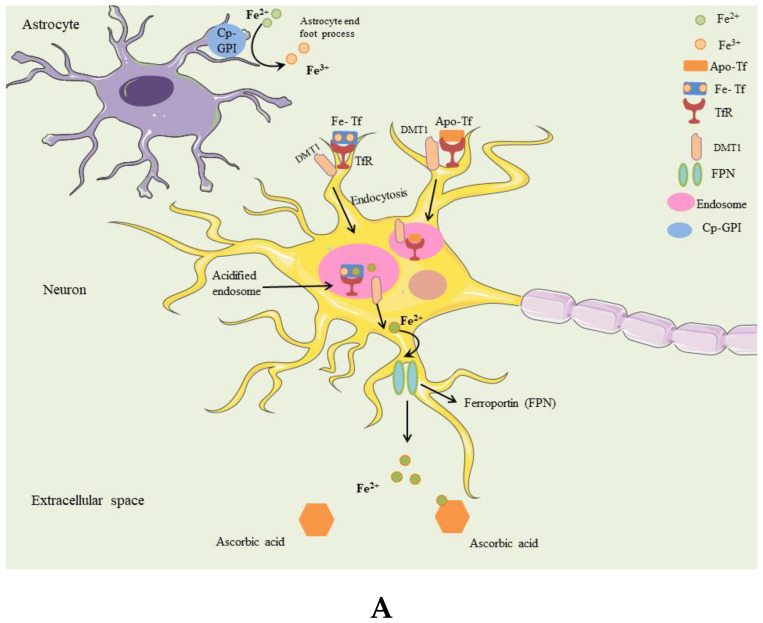
Metabolism of iron in the central nervous system. (**A**): Fe^3+^ is transported in the brain capillaries by transferrin. The brain capillary endothelial cells (BCEC) form the Blood-Brain-Barrier (BBB). On the luminal surface of the BCEC, the Transferrin R1 receptor (TfR1) captures the iron-loaded transferrin, and the transferrin/TfR1 is then internalized into an endosome. The endosome moves through the intracellular space and reaches the abluminal side of the BCEC. After that, the endosome fuses with the outer membrane. This exposes Fe^3+^ that is released into the interstitial space. Apo-transferrin still binds the TfR1 and undergoes endocytosis again to return to the luminal side BCEC. At this point, the apo-Tf is released. (**B**): The complex of Fe^3+^ transferrin travels through the extracellular space and reaches a neuron. The TfR1 on the neuron surface begin the endocytosis. Neuronal membrane exhibits DMT1 at difference from the BCEC. This wraps around the endosome and enters with it. Presynaptic vesicles bear ferroportin (not shown), suggesting that Fe^2+^ can travel via the synaptic vesicles to the inter-synaptic space, where Fe^2+^ is released after vesicle fusion. Thus, there is free iron in the pre- and postsynaptic cleft. On the outside of the tip of astrocytes, Glycosylphosphatidylinositol (GPI)-anchored ceruloplasmin (Cp-GPI) facilitates the oxidation of Fe^2+^ to Fe^3+^, allowing iron uploading into apo-transferrin. The intracellular and extracellular mechanisms can work synergistically to transfer iron to the neuronal cells.

**Figure 4 biomolecules-12-01248-f004:**
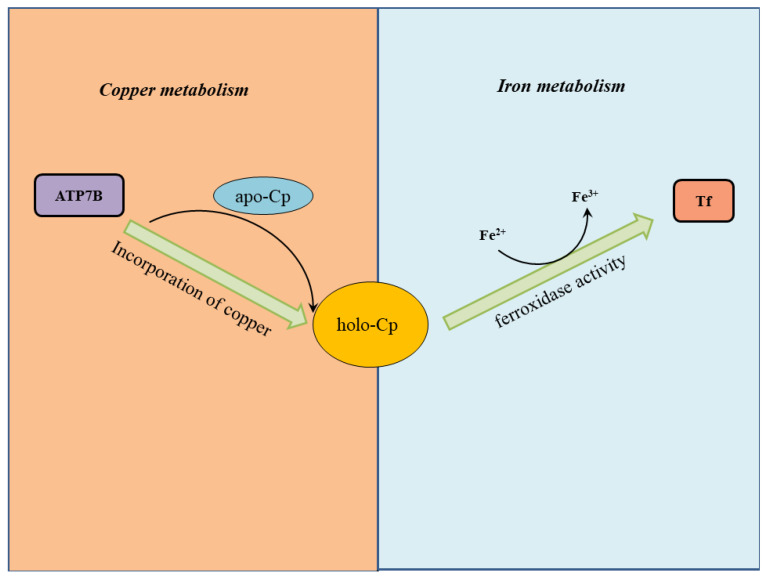
Ceruloplasmin is a crosstalk protein between iron and copper trafficking. Holo-ceruloplasmin (holo-Cp) is illustrated in yellow as a crosstalk protein that links iron and copper metabolism. The ATP7B copper pump in the hepatocyte uploads six copper atoms into apo-ceruloplasmin (apo-Cp). This allows the correct holo-Cp folding necessary for its ferroxidase activity: oxidation of Fe^2+^ to Fe^3+^ which can be then transfer to transferrin (Tf).

**Figure 5 biomolecules-12-01248-f005:**
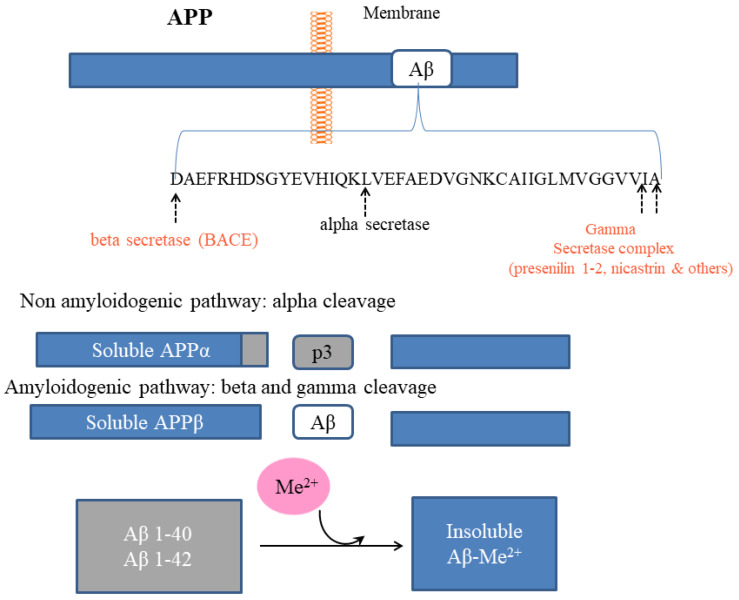
Pathway of Aβ production: the link with metals. The main constituent of amyloid aggregates is Aβ, a peptide of 39-43 amino acids, resulting from the cleavage of APP a transmembrane protein. More specifically, it has been reported that APP can be processed into two separate pathways. In the amyloidogenic pathway, APP is first cleaved by β-secretase at a site outside the cell surface. This generates a C-terminal fragment that is subsequently cleaved by γ-secretase at a site within the transmembrane domain. The product of this cleavage is Aβ. In the non-amyloidogenic pathway, APP is initially cut by α-secretase and then by γ-secretase, releasing a truncated P3 fragment. It was found that APP has specific copper and zinc-binding domains that catalyze redox activity, determining Aβ precipitation in plaques even at minimal concentrations. Copper and zinc domain in the Aβ peptide bind equimolar amounts of the two metals (Me^2+^). However, under acidosis conditions, Aβ zinc is copper completely displaced by copper.

**Table 1 biomolecules-12-01248-t001:** Iron absorption and distribution.

ENTEROCYTE	IRON	
	Enzyme/Protein	Protein Function
**Cell apical epithelium**	**non heme** CYBRD1 cytochrome b558 ferric/cupric reductase Steap2	Redox reduction of Fe^3+^ to Fe ^2+^
**Cell Membrane**	**non-heme** DMT-1	Channelling of Fe^2+^ via electrochemical gradient
	**heme** HCP-1	Import of heme iron
**Cytosol**	**non-heme** ferritin	Storage
	**heme** heme oxygenase	Cutting part of heme
**Cell basolateral membrane**	Ferroportin	Fe^2+^ is transported from the cytosol to the blood circulation
**Cell basolateral outer epithelium**	Hephaestin	Redox oxidation of Fe^2+^ to Fe^3+^
**Control**	Peptide hormone hepcidin	Iron excess: hepcidin controls ferroportin degradation and iron storage.
**Plasma**	Transferrin (holo/apo Tf)	Tf binds and transports 2 atoms of Fe^3+^
	Hemoglobin	Erythropoiesis

**Table 2 biomolecules-12-01248-t002:** Iron metabolism in the liver.

Hepatocyte	Iron	
	Enzyme/Protein	Protein Function
**Apical surface**	TfR1 Holo-Tf	Fe forms complexes with TfR1/Holo-Tf for internalization: Biosynthesis of acidified endosome
**Cytosol**	Reductases DMT1	Through an electrochemical gradient Fe^3+^ is reduced to to Fe^2+^ Channelling of Fe^2+^ outside of endosome
	Ferritin Citrate, AMP, ATP	Storage of Labile pool
**Interface with bile canaliculus**		
**Basolateral outer membrane**	Ferroportin	Fe^2+^ is transported from the cytosol to blood circulation
	Ceruloplasmin	Fe^2+^ is oxidized to Fe^3+^
**Control**	IRE/IRP	Iron is uptaken into the cell for utilization, storage, and transport
**Circulation**	Transferrin (holo/apo Tf)	Binding and transport of 2 atoms of Fe^3+^
	Ferritin	Iron storage protein

**Table 3 biomolecules-12-01248-t003:** Shared traits of inherited metal-metabolic diseases and neurodegenerative disorders.

Metal	Gene	Monogenic Disorder Linked to Iron Metabolism	Neurodegenerative Diseases with a Complex Etiology Linked to Iron Metabolism
	*CP*	Aceruloplasminemia	Parkinson’s Disease
Iron	*HFE*	Hereditary Hemochromatosis	Alzheimer’s Disease Parkinson’s Disease Amyloid Lateral Sclerosis
	*Ferritin*	H-Ferritin Related Iron Overload	-
	*TF*	Atransferrinemia	Alzheimer’s Disease
	*DMT-1*	DMT-1 deficiency	Alzheimer’s Disease Amyloid Lateral Sclerosis
	*IREB2*	-	Alzheimer’s Disease Parkinson’s Disease
	*TFR*	Non-HFE hereditary hemochromatosis	-

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
