# Peer review of "Iron in Alzheimer’s Disease: From Physiology to Disease Disabilities"

_biomolecules, 2022, doi:10.3390/biom12091248_

Round 1
Reviewer 1 Report
The review article by Pal el al. explains the basics of iron homeostasis and implications of its dysfunctional homeostasis in neurological conditions. The review has some deficiencies in its current format. Some of these are listed below:
- Some concepts are incompletely explained, for example the functional role of ceruloplasmin and hephaestin, incomplete listing of various ferrireductases, no mention of the role of iron in prion disorders especially in the section titled ‘Metals in neurological disorders’, and the role of cytokines in AD, especially via hepcidin.
- The diagrammatic representation of the various transport mechanisms is very sketchy and incomplete.
- Several important publications in the field of iron homeostasis are missing.
- There are several typos.
- Non-conventional abbreviation are used in many places, e.g RE instead of ER for endoplasmic reticulum.
- Diagrammatic representation of the BBB is too simplistic.
Author Response
Comments and Suggestions for Authors
The review article by Pal el al. explains the basics of iron homeostasis and implications of its dysfunctional homeostasis in neurological conditions. The review has some deficiencies in its current format. Some of these are listed below:
- Some concepts are incompletely explained, for example the functional role of ceruloplasmin and hephaestin, incomplete listing of various ferrireductases, no mention of the role of iron in prion disorders especially in the section titled ‘Metals in neurological disorders’, and the role of cytokines in AD, especially via hepcidin.
Response: as suggested, we added new paragraphs reporting the role of iron in cytokines, including information on Hepcidin (page 3, 5 and page 13). A comment on PrP has been reported in page 5. The text changed is in red font.
- The diagrammatic representation of the various transport mechanisms is very sketchy and incomplete.
Response: The diagrammatic representation of the various transport mechanisms
has been improved as per the suggestion.
- Several important publications in the field of iron homeostasis are missing.
Response: Authors have added seminal publications in the Fe homeostasis.
- There are several typos.
Response: Authors have revised the full manuscript for typo, spelling and grammatical errors.
- Non-conventional abbreviation are used in many places, e.g RE instead of ER for endoplasmic reticulum.
Response: All the Abbreviations have been re-checked, and all non -conventional abbreviations were corrected including that of ER.
- Diagrammatic representation of the BBB is too simplistic.
Response: we have modified the diagrammatic representation of the BBB.
Reviewer 2 Report
Amit Pal et al submitted the manuscript entitled "Iron in Alzheimer's disease: from physiology to disease disabilities" which described the role of iron dysregulation in Alzheimer's Disease. This is a well-written manuscript which will gain readers interest by providing state-of-the-art information for the metal related pathogenic mechanisms of Alzheimer's disease and other neurodegeneration.
Apart from the pathogenic role of iron accumulation in AD, it would be appreciated if authors can provide additional information/section on the current perspectives of the therapeutic options targeting metal accumulation such as metal chelation in AD.
Also, some grammatical and spelling mistakes are found throughout the manuscript that authors may need to do the proofreading.
Author Response
Comments and Suggestions for Authors
Amit Pal et al submitted the manuscript entitled "Iron in Alzheimer's disease: from physiology to disease disabilities" which described the role of iron dysregulation in Alzheimer's Disease. This is a well-written manuscript which will gain readers interest by providing state-of-the-art information for the metal related pathogenic mechanisms of Alzheimer's disease and other neurodegeneration.
Response: Authors welcome the encouraging comments of the reviewer.
Apart from the pathogenic role of iron accumulation in AD, it would be appreciated if authors can provide additional information/section on the current perspectives of the therapeutic options targeting metal accumulation such as metal chelation in AD.
Response: As per the reviewer section, a new section 4 entitled, “Current perspectives of the therapeutic options targeting iron chelation in AD” has been added in the manuscript. Mainly, Fe chelation has been discussed to avoid deviating from the theme of the manuscript.
Also, some grammatical and spelling mistakes are found throughout the manuscript that authors may need to do the proofreading.
Response: Authors have revised the full manuscript for typo, spelling and grammatical errors.
Round 2
Reviewer 1 Report
The authors have addressed my concerns appropriately.